# Seropositivity of SARS-CoV-2 in the Population of Kazakhstan: A Nationwide Laboratory-Based Surveillance

**DOI:** 10.3390/ijerph19042263

**Published:** 2022-02-17

**Authors:** Yuliya Semenova, Zhanna Kalmatayeva, Ainash Oshibayeva, Saltanat Mamyrbekova, Aynura Kudirbekova, Ardak Nurbakyt, Ardak Baizhaxynova, Paolo Colet, Natalya Glushkova, Alexandr Ivankov, Antonio Sarria-Santamera

**Affiliations:** 1Department of Neurology, Ophthalmology and Otolaryngology, Semey Medical University, Semey 071400, Kazakhstan; yumsem@mail.ru; 2School of Medicine, Al-Farabi Kazakh National University, Almaty 050040, Kazakhstan; Zhanna.Kalmatayeva@kaznu.kz (Z.K.); saltanat.mamyrbekova@kaznu.kz (S.M.); glushkovanatalyae@gmail.com (N.G.); 3Administrative Office, Khoja Akhmet Yassawi International Kazakh-Turkish University, Turkestan 161204, Kazakhstan; ainash.oshibayeva@ayu.edu.kz; 4Invitro-Kazakhstan Laboratory, Medical Department, Almaty 050000, Kazakhstan; akudirbekova@invitro.ru; 5Department of Public Health, Asfendiyarov Kazakh National Medical University, Almaty 050012, Kazakhstan; ardaknur@mail.ru; 6Department of Medicine, Nazarbayev University School of Medicine, Nur-Sultan 020000, Kazakhstan; ardak.baizhaxynova@alumni.nu.edu.kz (A.B.); paolo.colet@nu.edu.kz (P.C.); 7Independent Researcher, Almaty 050000, Kazakhstan; sash.ivankov@gmail.com

**Keywords:** COVID-19, Kazakhstan, seropositivity, surveillance

## Abstract

The data on seroprevalence of anti-SARS-CoV-2 antibodies in Kazakhstani population are non-existent, but are needed for planning of public health interventions targeted to COVID-19 containment. The aim of the study was to estimate the seropositivity of SARS-CoV-2 infection in the Kazakhstani population from 2020 to 2021. We relied on the data obtained from the results from “IN VITRO” laboratories of enzyme-linked immunosorbent assays for class G immunoglobulins (IgG) and class M (IgM) to SARS-CoV-2. The association of COVID-19 seropositivity was assessed in relation to age, gender, and region of residence. Additionally, we related the monitoring of longitudinal seropositivity with COVID-19 statistics obtained from Our World in Data. The total numbers of tests were 68,732 for SARS-CoV-2 IgM and 85,346 for IgG, of which 22% and 63% were positive, respectively. The highest rates of positive anti-SARS-CoV-2 IgM results were seen during July/August 2020. The rate of IgM seropositivity was the lowest on 25 October 2020 (2%). The lowest daily rate of anti-SARS-CoV-2 IgG was 17% (13 December 2020), while the peak of IgG seropositivity was seen on 6 June 2021 (84%). A longitudinal serological study should be envisaged to facilitate understanding of the dynamics of the epidemiological situation and to forecast future scenarios.

## 1. Introduction

The coronavirus (COVID-19) pandemic is having a profound impact across the globe. Infection with SARS-CoV-2 stimulates a detectable antibody response in most people. Antibody tests may help understand how many people have been infected with the virus that causes COVID-19. Serological tests identify infection status and were in high demand from the beginning of the COVID-19 pandemic due to the initial shortage of diagnostic tests that were coupled with asymptomatic infections. Nevertheless, public health authorities were hesitant to introduce them because of epidemiological and clinical uncertainties.

The Republic of Kazakhstan (hereafter, Kazakhstan) is the largest country of the Central Asian region, sharing borders with the Russian Federation in the north and China, Kyrgyzstan, Uzbekistan, and Turkmenistan in the south. Historically, Kazakhstan was closely bound to the Silk Road, lying at the intersection of population flows. So, when the coronavirus disease 2019 (COVID-19) emerged in China at the end of 2019, it was obvious that it would rapidly spread to Kazakhstan. However, the country has a relatively small population size and one of the lowest population densities in the world, which has, to a certain extent, constrained the transmission of infection [1].

In Kazakhstan, the first case of COVID-19 was officially registered on 13 March 2020; by March 16, the country’s government introduced a state of emergency that lasted until 11 May 2020 [2]. COVID-19 began to escalate immediately after the termination of lockdown measures imposed during that period, the consequence of which was a significant first wave from the end of June/early July to late August 2020 [3]. From November 2020 to February 2021, the number of cases remained relatively stable. Beginning in April 2021, a relevant surge in cases was observed, which represented the third wave, with the growth potentially associated with the Nauriz celebrations in the country in March. This wave was followed almost immediately by a fourth wave that began in June 2021 and that can be associated with the presence of the Delta variant in Kazakhstan [4].

Shortly after the first cases of COVID-19 were identified, the Kazakhstani government implemented extraordinary measures to control the spread of infection. A nationwide lockdown was imposed during the first wave, including the closure of borders and main cities, shutdowns of various services, including outpatient facilities and educational establishments, and nighttime curfews. Still, these measures were less stringent during the following waves, partly due to public protests. In addition to community-wide protection measures, it was also encouraged to follow individual protection measures, including social distancing, wearing of masks in public settings, frequent hand washing, and self-isolation [5].

Since the beginning of February 2021, Kazakhstan has implementing an anti-COVID-19 mass vaccination program and four vaccines are available to the public for free. Sputnik-V, an adenoviral vector-based vaccine [6], is the most frequently used in the country. In addition, three inactivated whole-virion vaccines have also been registered in Kazakhstan, one of them was locally developed: QazCovid-in^®^ [7]; the remaining two are HayatVax and Sinovac [8]. Despite unlimited access to anti-COVID-19 vaccination, the country has low vaccination rates, which can be explained by public fears [9].

The diagnosis of a COVID-19 case is based on real-time polymerase chain reaction (RT-PCR). In the case of a negative RT-PCR test but chest computed tomography scan findings compatible with COVID-19, a diagnosis of PCR-negative COVID-19 pneumonia is made (CITA). Serology tests are not officially used to confirm or rule out the diagnosis [10], but have been extensively used in the country both the identify acute infections when PCR tests were not available or to determine previous exposure to the virus. Thus, this study aimed to provide estimates on the seropositivity of SARS-CoV-2 infection in the Kazakhstani population over a one-year period and to investigate seropositivity in relation to the reporting of PCR-positive COVID-19 cases in such a period. We hope that this nationwide serological surveillance will assist policymakers in the planning of public health interventions targeted to COVID-19 containment.

## 2. Materials and Methods

### 2.1. Study Setting

Kazakhstan is administratively divided into 14 regions, which are further subdivided into 177 districts. The country has three cities of “republican significance”, which are separate administrative units: Nur-Sultan, the nation’s capital, previously known as Astana; Almaty, the former nation’s capital, previously known as Alma-Ata; and Shymkent, the third largest city in Kazakhstan. In Kazakhstan, the town district is the smallest urban administrative unit and the village is the smallest rural administrative unit. According to current estimates, the country’s population is around 19 million people, and the population density is 6 people per square kilometer, with the majority of inhabitants residing in urban areas (56%). The country’s economy is mainly dependent on mining, oil, and gas extraction [11].

Healthcare is mostly delivered via a network of government-owned health facilities, but there is also a significant private provision of healthcare services, which is often implemented through government private partnership or payments from patients. As for laboratory services, these are commonly provided through the well-developed net of private laboratories represented in every smallest administrative unit.

### 2.2. Study Design and Population

We relied on the data obtained from “IN VITRO” laboratories, the largest private network of government-licensed laboratories in Kazakhstan, accounting for approximately 70% of market share in clinical diagnostic testing. “IN VITRO” laboratories are available in all administrative units of Kazakhstan, including rural areas. Beginning in July 2020, the “IN VITRO” laboratories performed ELISA (enzyme-linked immunosorbent assay) testing of class G immunoglobulins (IgG) and class M (IgM) to SARS-CoV-2 (VectorBest, Novosibirsk, Russian Federation). A two-stage indirect ELISA test was used; the first step involved binding of IgG/IgM to the recombinant SARS-CoV-2 receptor-binding domain of spike glycoprotein (S-protein). The second stage involved interaction between the anti-human anti-IgG/IgM monoclonal antibodies (horseradish peroxidase) and IgG/IgM complexes. This immunoassay employs a signal/cutoff (S/CO) ratio of 1.1 for positivity, while a S/CO of less than 1.1 is reported as negative. The immunoassay detects IgG antibodies to the spike protein of SARS-CoV-2 with 90% sensitivity (95% confidence interval (CI): 76.4–96.4) and 100% specificity [12]. We obtained the results of all anti-SARS-CoV-2 ELISA IgM and IgG tests performed by the laboratory between 15 July 2020 and 10 July 2021 (12 months). Overall, our study comprised 68,722 test results.

### 2.3. Covariates

The association of COVID-19 seropositivity was assessed by age and gender. Age groups were categorized as follows: 0–9, 10–19, 20–29, 30–39, 40–49, 50–59, 60–69, 70–79, and ≥80 years old; gender was categorized as male and female. The seropositivity rates were evaluated during the four peak weeks: 3–9 August 2020 (week 32, 2020), 1–7 February 2021 (week 5, 2021), 26 April–2 May 2021 (week 17, 2021), 5–11 July 2021 (week 27, 2021). The other variables included in this analysis were obtained from Our World in Data COVID-19 dataset [13]: number of new cases, share of positive PCR tests, and number of vaccinated people.

### 2.4. Ethics Statement

We applied to the Ethics Committee of Semey Medical University for permission to perform the study (Protocol #9.1, dated 13 May 2021). Since the study team solely relied on anonymized medical records, the Ethics Committee waived the need for informed consent.

### 2.5. Statistical Analyses

The statistical analyses were processed via IBM SPSS Statistics software, version 20 (Chicago, IL, USA) for Windows. As a first step, we tested the type of data distribution and assessed the descriptive statistics of numerical variables. Qualitative variables were presented as frequencies and percentages. Relative risks and 95% confidence intervals (95% CI) were used to estimate relatively higher risk groups and the true percentages. The critical value of study variables was considered significant at *p* < 0.05.

## 3. Results

The total numbers of tests were 68,732 for SARS-CoV-2 IgM and 85,346 for IgG, of which 22% and 63% were positive, respectively. Table 1 and Table 2 depict the seropositivity of anti-SARS-CoV-2 IgM and IgG in the participating population during the four peak weeks of 2020–2021. Males had higher rate of positive anti-SARS-CoV-2 IgM results than females: the overall seropositivity constituted 23.7% (95% CI: 23.2%; 24.2%) versus 20.2% (95% CI: 19.8%; 20 6%), respectively. People aged 60 years and older had positive IgM results more often than younger age groups. For anti-SARS-CoV-2 IgG, the rate of positive results was comparable between males and females: 62.8% (95% CI: 62.3–63.3%) versus 63.1% (95% CI: 62.7–63.5%). Likewise, different age groups tested positive for anti-SARS-CoV-2 IgG in roughly equal proportions.

Figure 1 presents the data on seropositivity of anti-SARS-CoV-2 IgM and IgG in relation to COVID-19 statistics. Figure 2 reports the proportion of positive anti-SARS-CoV-2 IgM and IgG tests and vaccinated population. Data from the first months included in this work (July/August 2020) were characterized by the highest rate of positive anti-SARS-CoV-2 IgM results, which peaked on 16 July 2020, reaching a level of 53%. The rate of IgM seropositivity was the lowest on 25 October 2020 (2%). In general, there was a high agreement between the seropositivity of anti-SARS-CoV-2 IgM, daily new cases, and the prevalence of positive PCR tests during the entire period analyzed in this study (Figure 1A,B). The peak of IgG seropositivity was seen on 6 June 2021, when it was as high as 84%, while the nadir was observed on 13 December 2020 (17%). The rates of anti-COVID-19 vaccination were remarkably low; only 21.7% of the country’s population was fully vaccinated by the end of our study (Figure 2).

Figure 3 and Figure 4 reflect the estimated relative risks (RR) of IgM and IgG seropositivity in relation to the peak week and age. As compared with the reference week (24–30 January 2021), IgM seropositivity was the highest in the week of 2–8 August 2020, with a two-fold increase in RR. People aged 60 years and older had a more than two-fold increase in RR as compared with individuals aged 20–29 years. The risk of testing positive for IgG was the highest during the week of 27 June–3 July 2021, when the highest number of COVID-19 cases was reported. As compared with individuals aged 20–29 years, the risk of testing positive for anti-SARS-CoV-2 IgG was lower in 0–9, 10–19 and 30–39 age groups.

## 4. Discussion

This study has provided estimates of SARS-CoV-2 seropositivity in the Kazakhstani population over one year. We hope that this nationwide serological assessment may assist policymakers to understand the dynamics of the pandemic and therefore in planning public health interventions targeted to the containment of the COVID-19 pandemic. Surveillance is one of the key pillars to combat the COVID-19 pandemic. Resource-rich countries have developed extensive national and regional surveillance programs to monitor seroprevalence patterns over time. However, many countries lack both infrastructure and resources to implement such intensive surveillance programs.

The major limitation of this study is that it relies on self-referrals by individuals, although it is based on a substantial sample and covers a prolonged period of observation. Another critical limitation comes from the fact that our data are based on the test and not on individuals who were tested. Thus, a single person may have been included more than once in this analysis. In addition, the period of observation ends in early July 2021. In the following months, the country was affected by the wave caused by the Delta variant that started in June and reached its peak in mid-August. The data on testing and positivity during that wave were not available for inclusion in this work.

However, the data presented here have certain strengths. To the best of our knowledge, data on the seroprevalence of anti-SARS-CoV-2 antibodies in the Kazakhstani population are non-existent, but are needed to construct action plans both at the local and national levels.

The daily prevalence of positive IgM tests was higher at the end of July and the start of August 2020 as well as during April–May 2021. This effect may be attributed to the lack of access to PCR testing during those critical moments, probably reflecting that IgM testing was used to diagnose the disease. The antibodies elicited by vaccination, which started slowly in the country in February 2021, may have contributed to the increase in IgG-positive cases that were observed after April 2021. Vaccination, nevertheless, started slowly and only showed an increase after the March–April 2021 wave.

In general, the proportion of individuals who tested positive for IgM was lower than that of individuals who tested positive for IgG. We assume that this could be explained by the period when both tests are positive. IgM positivity is found during the acute phase and disappears after 4–5 weeks. IgG is positive after 2–3 weeks and may remain detectable after 6 months. Because self-isolation was mandatory for symptomatic cases, many COVID-19 patients may have serological testing later. A significant proportion of COVID-19 cases is asymptomatic, and there may be a reasonable number of individuals who tested positive for anti-SARS-CoV-2 IgG belonging to this large group of asymptomatic cases.

Unfortunately, no data are publicly available in the country describing the infection rates by age and sex, so it is not possible to determine to what extent those positive percentages may reflect the age–sex composition of COVID-19 infection in Kazakhstan. However, it is interesting to note that individuals aged 70 years and older had a higher likelihood of a positive IgM test, although the proportion of IgG-positives was not higher among them. It is impossible to conclude what was the origin of those positive antibodies—infection or vaccination—and whether they reflect a shorter duration of immunological response in individuals of advanced age with the available data.

Several international studies have reported the rates of anti-SARS-CoV-2 antibodies in different population groups. For example, a seroprevalence study that used sera collected between 28 September and 30 December 2020, in the Chelyabinsk region of the Russian Federation, showed that the crude prevalence of anti-SARS-CoV-2 Spike IgM was 3.2% [14]. Another seroprevalence study performed on the population of 26 regions of the Russian Federation identified that 17.8% of individuals had IgG antibodies to SARS-CoV-2 [15]. Another seroprevalence study performed on the population of 26 regions of the Russian Federation identified that 17.8% of individuals had IgG antibodies to SARS-CoV-2 [16]. When interpreting the results of seroprevalence studies, it should be taken into consideration that specific populations were tested; thus, their results may not be directly comparable.

Many countries introduced serological surveillance of anti-SARS-CoV-2 antibodies among the general population to assess the prevalence of COVID-19. Quite often, these programs relied on repetitive population-based surveys conducted at predetermined intervals analyzing samples that have population representativeness [17]. When adequately planned, such surveys enable the accurate estimates of COVID-19 prevalence with minimum bias since antibody testing allows determination of exposure to SARS-CoV-2, thereby providing a full picture of the whole impact of the pandemic. PCR tests are valuable characteristics for diagnosing cases, but when they are not available or in asymptomatic cases, there are obvious limitations. Thus, serological tests provide a clearer picture of COVID-19 spread. Repeating the tests over time help to reveal changes in the magnitude of this infection [18].

In our study, the lowest daily rate of anti-SARS-CoV-2 IgG was 17%, but since early March 2021 more than 50% of tests performed have yielded positive results. This finding raises the issue of COVID-19 herd immunity, which has been strongly debated over the last year. Initially, it was considered that when a “herd-immunity threshold” is reached, the pandemic will end. Different suggestions on the desirable proportion of the population with anti-SARS-CoV-2 antibodies were obtained either through exposure to SARS-CoV-2 or vaccination [19]. However, the mutations in the virus made it obvious that the proportion of the population required to reach a herd-immunity threshold is not a “fixed value”, due to a variety of reasons, some associated with the emergence of new SARS-CoV-2 variants, including waning immunity from previous infections or vaccination, but also because of unequal vaccination coverage rates, the number of doses, and vaccine hesitancy. Moreover, it is currently thought that COVID-19 will eventually become an endemic disease, very much like influenza [20].

The emergence of the B.1.617.2 (Delta) variant in Southeast Asia changed the epidemiology of COVID-19 disease. This variant is transmitted more easily and thus requires a higher threshold for herd immunity [21]. According to Our World in Data, as of 31 August 2021, the share of the Delta variant has reached almost 100% in many countries [13]. In this regard, it might be supposed that the dramatic increase in the incidence of COVID-19 in Kazakhstan that started soon after June 2021 [22]. In this regard, it might be supposed that the dramatic increase in the incidence of COVID-19 in Kazakhstan that started soon afterward after June 2021 [4] could be explained by the spread of B.1.617.2 variant across the country.

Although SARS-CoV-2 is constantly mutating and novel variants emerge, our study could help to make the nationwide policy to manage COVID-19. Because there is a cross-reactive antibody response between wild-type SARS-CoV-2 and novel coronavirus variants identified to date, our data could be used to estimate the level of population immunity achieved thus far. Most likely, the application of such community-wide mitigation measures such as quarantine is no longer needed, although compliance with an individual protection and hygiene concept should be encouraged. Besides, there is a need to achieve high vaccination coverage, especially among individuals with high-risk conditions, and to envisage immunological surveillance to monitor population immunity, which may also help to clarify the issue of booster shots, including the timing and priority vulnerable groups.

Most countries have primarily focused on the most vulnerable population groups when rolling out their anti-COVID-19 vaccination programs, including frontline workers, the elderly, and people with chronic diseases. Kazakhstan was no exception; the first group targeted was frontline healthcare workers, beginning mid-February 2021, and vaccination of the public started in April 2021. Although vaccination is free and broad geographic coverage is enabled, less than one-third of the population is fully vaccinated, mainly due to high rates of vaccine hesitancy [9]. Vaccination may, however, have contributed to the increase in positive IgG tests identified after April 2021. The lack of specificity of the test, which can detect IgG either natural immunity or COVID-19 vaccination, makes it impossible to determine their origin.

Still, because of the continuous appearance of novel SARS-CoV-2 variants that may escape the vaccine-induced response and waning post-vaccination immunity, COVID-19 may resurge, even in well-vaccinated countries [23]. International efforts should be implemented to analyze the seroprevalence against SARS-CoV-2 after vaccination or natural infection and how it is transformed to prevented infections and deaths.

## 5. Conclusions

The findings of our study should be interpreted with caution since individuals who opted for serological testing are likely to differ from the general population in terms of exposure to SARS-CoV-2. However, the nationwide coverage of the data along with the longitudinal design enable promising insights, which may be of interest to policymakers. The major finding is that around three-fourths of the participants in this study have developed antibodies against SARS-CoV-2. However, this estimate is likely to be exaggerated at the population level. Even supposing this statement to be correct, public health measures should be sustained to constrain the spread of novel viral variants with mutations that enable significant immune-evading capacities. In addition, a longitudinal serological study with appropriate sampling and population representativeness should be envisaged to facilitate understanding of the current epidemiological situation and to forecast future scenarios.

## Figures and Tables

**Figure 1 ijerph-19-02263-f001:**
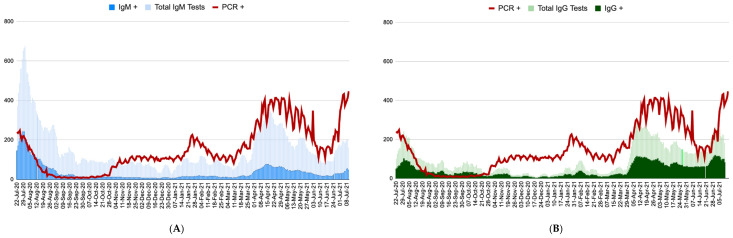
Daily prevalence of PCR-positive tests in the period from 16 July 2020 to 7 July 2021 versus (**A**) daily anti-SARS-CoV-2 IgM tests and positive results and (**B**) daily anti-SARS-CoV-2 IgG and positive results.

**Figure 2 ijerph-19-02263-f002:**
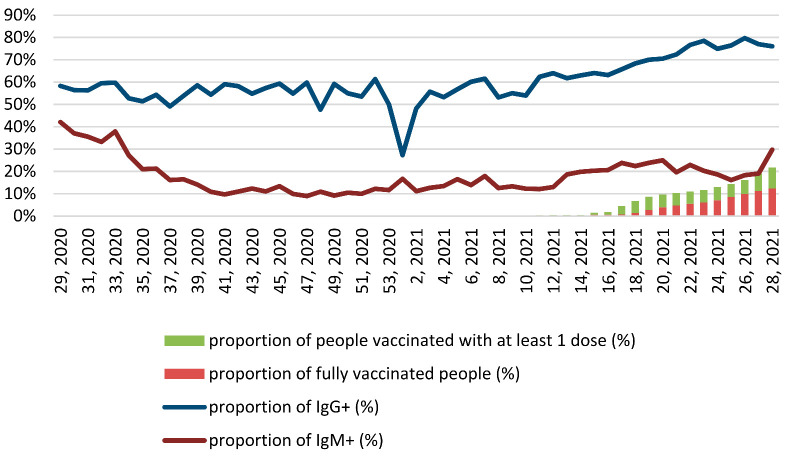
Weekly proportion of positive anti-SARS-CoV-2 IgM and IgG tests and vaccinated population.

**Figure 3 ijerph-19-02263-f003:**
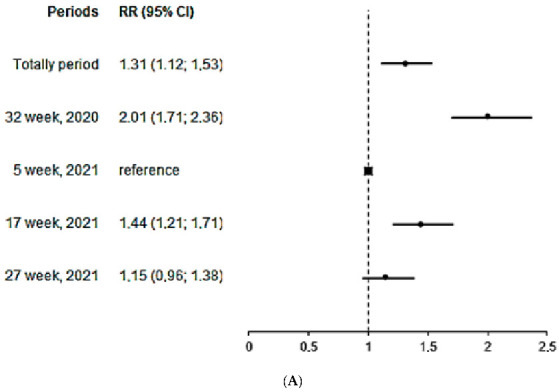
Estimated relative risk (RR) of IgM seropositivity by the peak week (**A**) and age (**B**).

**Figure 4 ijerph-19-02263-f004:**
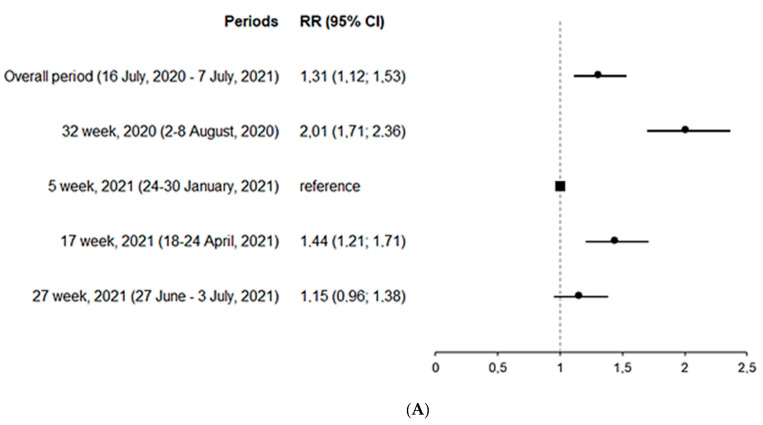
Estimated relative risk (RR) of IgG seropositivity by the peak week (**A**) and age (**B**).

**Table 1 ijerph-19-02263-t001:** Seropositivity of anti-SARS-CoV-2 IgM in the population of Kazakhstan during four peak weeks in 2020–2021.

Variables	Week 32, 2020 2–8 August 2020		Week 5, 2021 24–30 January 2021		Week 17, 2021 18–24 April 2021		Week 27, 2021 27 June–3 July 2021		Overall Seropositivity 16 July 2020–7 July 2021	
	*p* (%)	95% CI	*p* (%)	95% CI	*p* (%)	95% CI	*p* (%)	95% CI	*p* (%)	95% CI
Gender										
Male	32.9	30.4; 35.4	19.6	15.0; 24.2	28.6	25.7; 31.6	21.1	18.0; 24.2	23.7	23.2; 24.2
Female	33.5	31.2; 35.8	14.9	11.9; 17.9	20.8	18.7; 22.9	17.8	15.5; 20.1	20.2	19.8; 20.6
Age group, years										
0–9	10.9	2.7; 19.1	27.3	1.0; 53.6	51.7	33.5; 69.9	0.0	0.0; 0.0	15.5	13.1; 17.9
10–19	23.8	16.2; 31.3	22.2	8.6; 35.8	25.5	16.7; 34.3	15.8	7.6; 24.0	18.8	17.2; 20.3
20–29	22.1	18.2; 26.1	11.2	5.7; 16.7	15.4	11.4; 19.4	13.0	9.0; 17.0	15.1	14.5; 15.8
30–39	25.7	22.7; 28.7	14.7	10.4; 19.1	23.4	20.1; 26.6	18.8	15.6; 22.1	18.9	18.4; 19.5
40–49	32.1	28.6; 35.6	18.3	12.7; 24.0	24.1	20.5; 27.6	18.2	14.3; 22.0	22.3	21.6; 22.9
50–59	44.9	40.4; 49.3	15.6	9.3; 21.9	29.0	24.2; 33.7	24.9	19.3; 30.4	26.7	25.8; 27.5
60–69	55.3	49.8; 60.8	22.4	12.4; 32.4	26.0	20.3; 31.7	25.3	18.4; 32.3	30.6	29.4; 31.7
70–79	46.3	35.3; 57.2	26.3	6.5; 46.1	19.7	10.8; 28.7	29.8	16.7; 42.9	30.6	28.5; 32.7
>80	56.5	31.0; 73.7	20.0	−15.1; 55.1	15.0	−0.6; 30.6	0.0	0.0; 0.0	32.9	28.4; 37.3
Country region (city)										
Aktobe region	44.2	29.3; 59.0	0.0	0.0; 0.0	28.6	14.9; 42.2	32.3	15.8; 48.7	28.0	25.3; 30.7
Almaty city	33.7	31.4; 36.1	17.3	14.1; 20.4	24.1	22.1; 26.2	19.8	17.5; 22.1	21.5	21.1; 21.9
Almaty region	33.3	22.9; 43.8	4.8	−4.3; 13.9	25.5	16.7; 34.3	16.1	7.0; 25.3	22.9	21.2; 24.6
Atyrau region	36.6	30.6; 42.6	20.0	−15.1; 55.1	26.7	15.5; 37.9	24.2	9.6; 38.9	25.7	24.3; 27.2
East Kazakhstan region	31.5	25.1; 37.9	25.5	13.1; 38.0	17.1	4.7; 29.6	17.1	5.6; 28.6	23.2	21.9; 24.6
Karaganda region	32.6	18.6; 46.6	7.7	−6.8; 22.2	17.9	3.7; 32.0	23.1	11.6; 34.5	22.5	19.9; 25.0
Kyzylorda region	40.3	28.1; 52.5	0.0	0.0; 0.0	18.9	6.3; 31.5	11.1	−3.4; 25.6	13.2	11.9; 14.5
Mangystau region	51.9	33.0; 70.7	16.7	−13.2; 46.5	25.0	6.0; 44.0	19.0	8.9; 29.1	30.7	26.8; 34.5
Nur-Sultan city	26.7	22.7; 30.7	19.2	11.4; 26.9	20.3	14.8; 25.8	16.3	11.4; 21.2	20.7	19.8; 21.5
Pavlodar region	33.0	23.1; 42.8	12.5	−10.4; 35.4	100.0	100.0; 100.0	11.1	−3.4; 25.6	17.4	15.3; 19.5
Shymkent city	36.3	29.0; 43.6	0.0	0.0; 0.0	23.9	14.0; 33.9	9.5	0.6; 18.4	23.5	21.9; 25.1
Zhambyl region	28.6	4.9; 52.2	100.0	100.0; 100.0	100.0	100.0; 100.0	15.4	−4.2; 35.0	28.5	23.5; 33.5
Republic of Kazakhstan	33.2	31.5; 34.9	16.5	14.0; 19.1	23.8	22.1; 25.6	19.0	17.2; 20.9	21.7	21.4; 22.0

95% CI—95% Confidence Interval.

**Table 2 ijerph-19-02263-t002:** Seropositivity of anti-SARS-CoV-2 IgG in the population of Kazakhstan during four peak weeks in 2020–2021.

Variables	Week 32, 2020 3–9 August 2020		Week 5, 2021 1–7 February 2021		Week 17, 2021 26 April–2 May 2021		Week 27, 2021 5–11 July 2021		Overall Seropositivity 16 July 2020–7 July 2021	
	*p* (%)	95% CI	*p* (%)	95% CI	*p* (%)	95% CI	*p* (%)	95% CI	*p* (%)	95% CI
Gender										
Male	59.5	57.1–61.9	58.4	53.3–63.4	58.4	53.3–63.4	77.5	75.0–80.0	62.8	62.3–63.3
Female	59.5	57.3–61.8	55.6	51.4–59.9	55.6	51.4–59.9	76.7	74.8–78.6	63.1	62.7–63.5
Age group, years										
0–9	52.6	39.7; 65.6	30.0	1.6; 58.4	62.9	46.8; 78.9	64.0	45.2; 82.8	53.9	51.0; 56.9
10–19	58.1	49.8; 66.4	56.3	42.2; 70.3	57.0	48.5; 65.6	63.6	54.9; 72.2	59.0	57.3; 60.8
20–29	65.4	61.2; 69.6	54.0	45.3; 62.7	66.9	62.3; 71.5	76.0	71.8; 80.2	62.6	61.7; 63.5
30–39	54.4	51.2; 57.6	55.9	49.6; 62.2	64.9	61.6; 68.1	76.3	73.5; 79.1	61.0	60.4; 61.7
40–49	56.4	53.0; 59.9	56.1	49.3; 62.9	65.3	62.0; 68.6	77.9	74.8; 81.0	62.8	62.1; 63.5
50–59	63.1	59.0; 67.1	57.9	50.0; 65.7	72.3	68.5; 76.0	80.1	76.4; 83.8	65.6	64.8; 66.4
60–69	66.6	61.5; 71.6	56.0	45.3; 66.6	66.8	62.0; 71.5	80.7	76.4; 84.9	67.3	66.3; 68.3
70–79	65.6	56.1; 75.1	85.2	71.8; 98.6	52.3	43.7; 60.9	76.8	68.4; 85.1	64.4	62.6; 66.1
>80	62.1	44.4; 79.7	71.4	38.0; 104.9	51.7	33.5; 69.9	53.8	26.7; 80.9	59.7	55.8; 63.7
Country region (city)										
Aktobe region	77.8	66.7; 88.9	68.8	46.0; 91.5	72.0	61.8; 82.2	89.4	82.0; 96.8	75.4	73.2; 77.6
Almaty city	56.5	54.2; 58.8	54.6	50.6; 58.5	64.0	62.0; 65.9	76.7	74.9; 78.6	61.7	61.3; 62.1
Almaty region	54.5	44.1; 64.9	63.6	43.5; 83.7	73.8	66.3; 81.4	80.4	72.8; 87.9	59.4	57.6; 61.2
Atyrau region	62.5	56.7; 68.4	80.0	59.8; 100.2	64.3	56.1; 72.6	78.1	69.9; 86.4	67.9	66.6; 69.2
East Kazakhstan region	44.5	37.3; 51.7	82.4	64.2; 100.5	78.0	67.4; 88.5	76.2	67.1; 85.3	60.2	58.5; 61.8
Karaganda region	77.4	66.1; 88.6	60.0	35.2; 84.8	77.3	64.9; 89.7	82.8	74.8; 90.7	68.1	65.8; 70.5
Kyzylorda region	62.2	46.5; 77.8	57.1	38.8; 75.5	67.3	54.2; 80.5	87.5	74.3; 100.0	62.2	60.3; 64.2
Mangystau region	70.4	53.1; 87.6	50.0	10.0; 90.0	62.5	43.1; 81.9	48.5	36.4; 60.5	67.4	63.8; 71.0
Nur-Sultan city	64.4	60.7; 68.1	54.6	46.4; 62.8	66.8	61.4; 72.2	80.1	75.9; 84.4	63.6	62.7; 64.5
Pavlodar region	73.7	65.1; 82.4	69.2	44.1; 94.3	84.6	65.0; 104.2	74.1	57.5; 90.6	63.2	60.7; 65.7
Shymkent city	65.0	58.0; 72.0	68.4	47.5; 89.3	71.0	62.4; 79.6	72.0	61.8; 82.2	70.9	69.4; 72.4
Zhambyl region	55.6	32.6; 78.5	100.0	100.0; 100.0	86.4	72.0; 100.0	72.2	57.6; 86.9	67.1	63.3; 71.0
Republic of Kazakhstan	59.5	57.9; 61.2	56.7	53.5; 60.0	65.7	64.1; 67.4	77.0	75.5; 78.5	63.0	62.7; 63.3

95% CI—95% Confidence Interval.

## Data Availability

Data available on request.

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
