# Peer review of "Seropositivity of SARS-CoV-2 in the Population of Kazakhstan: A Nationwide Laboratory-Based Surveillance"

_ijerph, 2022, doi:10.3390/ijerph19042263_

Round 1

Reviewer 1 Report

This is an interesting topic, and have potential clinical conclusion for SARS-CoV-2 positivity and policymaking for vaccination and other COVID-19 managements. With some positive side this study has several limitations too like:

  1. Authors have reported only for a year data with one variant. Now other variants o of SARS-CoV-2 emerging. So how this data help to make a nationwide policy to help in management of SARS-CoV-2. I would encourage authors to discuss how this study helps to make policy to manage the SARS-CoV-2 in Kazakhstan.  
  2. Need rigorous English editing.
  3. Figures quality is very poor, please improve the figure quality.

Author Response

Reviewer 1

Comments for the author

Changes by the author

This is an interesting topic, and have potential clinical conclusion for SARS-CoV-2 positivity and policymaking for vaccination and other COVID-19 managements. With some positive side this study has several limitations too like:

Thank you evaluation of our manuscript and for your thoughtful approach.

We introduced the changes you have proposed and amended the corresponding parts of our manuscript.

We marked up all changes using the “Track
Changes” function.

1

Authors have reported only for a year data with one variant. Now other variants o of SARS-CoV-2 emerging. So how this data help to make a nationwide policy to help in management of SARS-CoV-2. I would encourage authors to discuss how this study helps to make policy to manage the SARS-CoV-2 in Kazakhstan.

Done. We added the following passage to the Discussion section:

Although SARS-CoV-2 is constantly mutating and novel variants emerge, our study could help to make a nationwide policy to manage COVID-19. Because there is a cross-reactive antibody response between wild-type SARS-CoV-2 and novel coronavirus variants identified to date, our data could be used to estimate the level of population immunity achieved thus far. Most likely, application of such communitywide mitigation measures as quarantine is no longer needed, while compliance with an individual protection and hygiene concept should be encouraged. Besides, there is a need to achieve high vaccination coverage, especially among individuals with high-risk conditions, and to envisage immunological surveillance to monitor population immunity, which may also help to clarify the issue of booster shots, including the timing and priority vulnerable groups.

2

Need rigorous English editing.

Done

3

Figures quality is very poor, please improve the figure quality.

Thank you. Done. We improved the quality of all figures.

Reviewer 2 Report

In this manuscript, Semenova et al studied seropositivity of SARS-CoV-2 infection in the Kazakhstani population from 2020 to 2021 using ELISA assays testing of IgG and IgM to SARS-CoV-2. And the authors also analyzed the association of COVID-19 seropositivity in relation to age, gender, and region. The study was based on a substantial number of samples and covered a prolonged period of observation and provided a certain useful information on SARS-CoV-2 seropositivity in the population of Kazakhstan. But as the authors mentioned, there were some potential limitations of this study: relying on referral of individuals, a single person included more than once, not including the delta wave in this analysis.

Minor comments

  1. line 26, “IgG” is missing following 85,346.
  2. Line 30, in the sentence “A longitudinal serological study with…” , delete “with” or revise
  3. Line 139, “IgG” is missing following 85,346.
  4. Line 151, “int his work” should be “in this work”.
  5. Figure 3B and Figure 4B are redundant.

Author Response

Reviewer 2

Comments for the author

Changes by the author

In this manuscript, Semenova et al studied seropositivity of SARS-CoV-2 infection in the Kazakhstani population from 2020 to 2021 using ELISA assays testing of IgG and IgM to SARSCoV-2. And the authors also analyzed the association of COVID19 seropositivity in relation to age, gender, and region. The study was based on a substantial number of samples and covered a prolonged period of observation and provided a certain useful information on SARS-CoV-2 seropositivity in the population of Kazakhstan. But as the authors mentioned, there were some potential limitations of this study: relying on referral of individuals, a single person included more than once, not including the delta wave in this analysis.

Thank you very much for reviewing our manuscript and for your valid comments.

We revised our manuscript accordingly. We marked up all changes using the “Track Changes” function.

1

line 26, “IgG” is missing following 85,346.

Done

The total number of tests was 68,732 SARS-CoV-2 IgM and 85,346 IgG, of which 22 and 63 % were positive, respectively.

2

Line 30, in the sentence “A longitudinal serological study with…” , delete “with” or revise

Done

A longitudinal serological study should be envisaged to facilitate understanding the dynamics of the epidemiological situation and to forecast future scenarios.

3

Line 139, “IgG” is missing following 85,346

Done

The total number of tests was 68,732 SARS-CoV-2 IgM and 85,346 IgG, of which 22 and 63 % were positive, respectively.

4

Line 151, “int his work” should be “in this work”.

Done

Data from the first months included in this work (July/August 2020) were characterized by the highest rate of positive anti-SARS-CoV-2 IgM results.

5

Figure 3B and Figure 4B are redundant.

Done. We have removed these figures.

Reviewer 3 Report

Line 109 - pls provide more details of the IgM assay from VectorBest? Is it to nucleocapsid or Spike? Reference 12 only covers Vector's IgG assay.

Line 113 statement about sensitivity & specificity applies to the Spike IgG assay only. Pls state this clearly.

Line 114 - states IgA (?should be IgM instead).

Line 139 - missing IgG in the sentence?

Line 150 - missing/incomplete sentence.

Line 151 - int his should read 'in this'

Line 200 - should read 'delta wave starting in June 2021' 

Paragraph in Line 206-210: where is the data that supports the conclusion that IgM positivity is higher early in 2020 & Mar-Apr 2021. The comments in this section are speculative.

Line 222 - IgM to which epitope (nucleocapsid or spike)

Line 224-226 - What antibodies - IgG or M or total?

Line255-259 - Cannot ascribe increased antibodies to Delta as they be also from the commencement of vaccination.

Line 260-300 - Disjointed ideas that deviate from the research topic at hand - CoV serology in Kazakhstan.

You are trying to relate antibody test results to overall prevalence data. You need individual patient data to individual patient PCR positivity. It might be better to concentrate on pre-vaccination phase alone or then compare it to the post-vaccination phase. And maybe distill out any regional differences in antibody/vaccination in Kazakhstan.

Pls explain why IgM titers are lower compared o IgG in all the columns in Table 1 & 2. 

Author Response

Reviewer 3

Comments for the author

Changes by the author

1

Line 109 - pls provide more details of the IgM assay from VectorBest? Is it to nucleocapsid or Spike? Reference 12 only covers Vector's IgG assay.

Thank you very much for reviewing our manuscript. Your comments helped us to improve the quality of our study and thus, made it better.  We marked up all changes using the “Track Changes” function.

Done.

A two-stage indirect ELISA test was used, the first step involved binding of IgG/IgM to the recombinant SARS-CoV-2 receptor-binding domain of spike glycoprotein (S-protein). The second stage involved interaction between the anti-human anti-IgG/IgM monoclonal antibodies (horseradish peroxidase) and IgG/IgM complexes.

2

Line 113 statement about sensitivity & specificity applies to the Spike IgG assay only. Pls state this clearly.

Done

The immunoassay detects IgG antibodies to the spike protein of SARS-CoV-2

3

Line 114 - states IgA (?should be IgM instead).

Done

We obtained the results of all anti-SARS-CoV-2 ELISA IgM and IgG tests performed by the laboratory from July 15, 2020, to July 10, 2021.

4

Line 139 - missing IgG in the sentence?

You are right! Done.

The total number of tests was 68,732 SARS-CoV-2 IgM and 85,346 IgG, of which 22 and 63 % were positive, respectively.

5

Line 150 - missing/incomplete sentence.

Done.

Figure 2 reports the proportion of positive anti-SARS-CoV-2 IgM and IgG tests and vaccinated population.

6

Line 151 - int his should read 'in this'

You are right.

Data from the first months included in this work (July/August 2020) were characterized by the highest rate of positive anti-SARS-CoV-2 IgM results.

7

Line 200 - should read 'delta wave starting in June 2021'

You are right again! Done.

Also, the period of observation ends in early July and since the country was affected by the wave caused by the Delta variant that started in June and reached its peak in mid-August.

8

Paragraph in Line 206-210: where is the data that supports the conclusion that IgM positivity is higher early in 2020 & Mar-Apr 2021. The comments in this section are speculative.

We rephrased this sentence to make it more specific and clarified further our statement:

The daily prevalence of positive IgM tests was higher at the end of July and the start of August 2020 as well as during April-May 2021. This effect may be attributed to the lack of access to PCR testing during those critical moments, probably reflecting that IgM testing was used to diagnose the disease.

9

Line 222 - IgM to which epitope (nucleocapsid or spike)

We made this statement more specific:

anti-SARS-CoV-2 Spike IgM

10

Line 224-226 - What antibodies - IgG or M or total?

We made this statement more specific:

IgG antibodies to SARS-CoV-2.

11

Line255-259 - Cannot ascribe increased antibodies to Delta as they be also from the commencement of vaccination

You are right. For this reason, we removed the following sentence:

This could contribute to a more intensive spread of the infection, as evidenced by the increase in the proportion of positive IgM tests during this period.

12

Line 260-300 - Disjointed ideas that deviate from the research topic at hand - CoV serology in Kazakhstan

You are trying to relate antibody test results to overall prevalence data. You need individual patient data to individual patient PCR positivity. It might be better to concentrate on pre-vaccination phase alone or then compare it to the post-vaccination phase. And maybe distill out any regional differences in antibody/vaccination in Kazakhstan

Yes, you are right. We condensed the ideas expressed in lines 260-300, leaving only those that are relevant for the study.

You are right again. Unfortunately, by the time we have accomplished the data collection, less than 10 % of the country’s population was fully vaccinated, which does not afford us the opportunity to fully evaluate this effect. As explained in the text, vaccination started in February in the country and was very slow, and only showed certain increase after May, when Delta was identified in the country and the June-August wave. At the same time, no proper regional data is publicly available in the country on vaccination status. For these reasons, we continue to show the data on seropositivity for the initial period considered in the study.

13

Pls explain why IgM titers are lower compared to IgG in all the columns in Table 1 & 2.

Done. We added the following passage to the Discussion section:

In general, the proportion of individuals who tested positive for IgM was lower than that of individuals with positive IgG. We assume that this could be explained by the period when both tests are positive. IgM positivity is found during the acute phase and disappear after 4-5 weeks. IgG is positive after 2-3 week and may remain detectable after 6 months. Because self-isolation was mandatory for symptomatic cases, many COVID-19 patients may have serological testing later. A significant proportion of COVID-19 cases are asymptomatic, and there may be a reasonable number of individuals who tested positive for anti-SARS-CoV-2 IgG belonging to this large group of asymptomatic cases.

Round 2

Reviewer 1 Report

Authors answers my comments and significantly improve the manuscript. I would recommend the current manuscript to publish in your journal.

Reviewer 3 Report

Line 119-121 only states the sensitivity & specificity of the IgG assay. What about the IgM's performance in this regard? Was IgG & IgM tested in all samples on the same occasion? 

It is implied in Section 2.2 that the Vector Best assay detects the IgG & IgM together in the same assay in the same analytical run. I don't think so. These are 2 separate tests. Please re-state clearly.  

You have not considered the fact that in this virus the antibody kinetics are such that IgG & IgM track each other closely i.e. they arise at the same time, or IgG earlier and lastly IgM arises earlier.  Thus lines 226-231 implies all the samples had both IgG & IgM tested.

Line 274 "afterward" should be deleted.

Line 293-295 To distinguish origin of the IgG - both nucleocapsid and spike antibodies can be measured to help clarify this. Sputnik vaccinees should have only spike Ab and no nucleocapsid Ab unless they have been infected with the virus in contrast to the whole virion vaccine (both present).

Line 304-305 Please elaborate how this research would help in policy formulation given the admixture of different vaccines and different variants at different time points.

Please put in the number of subjects beside the % for each category, column and row in Tables 1 & 2.

Line 197 - the study is not self-referrals by individuals but an analysis of test results performed by the lab which may contain multiple results for the same individual as received by the lab.

A major limitation is the contamination of the lab data through the use of different vaccines at different time points. Could the subjects be segregated by the same vaccine type and analysed separately?